# Epstein–Barr Virus in Salivary Samples from Systemic Lupus Erythematosus Patients with Oral Lesions

**DOI:** 10.3390/jcm10214995

**Published:** 2021-10-27

**Authors:** Alessio Buonavoglia, Patrizia Leone, Marcella Prete, Antonio Giovanni Solimando, Chiara Guastadisegno, Gianvito Lanave, Michele Camero, Vito Martella, Lorenzo Lo Muzio, Vito Racanelli

**Affiliations:** 1Unit of Internal Medicine “Guido Baccelli”, Department of Biomedical Sciences and Human Oncology, University of Bari Medical School, 70124 Bari, Italy; alessio.buonavoglia85@gmail.com (A.B.); patrizia.leone@uniba.it (P.L.); marcella.prete@uniba.it (M.P.); antonio.solimando@uniba.it (A.G.S.); chiara.guastadisegno@uniba.it (C.G.); vito.racanelli1@uniba.it (V.R.); 2IRCCS Istituto Tumori “Giovanni Paolo II”, 70124 Bari, Italy; 3Department of Veterinary Medicine, University of Bari, 70010 Valenzano, Italy; michele.camero@uniba.it (M.C.); vito.martella@uniba.it (V.M.); 4Department of Clinical and Experimental Medicine, University of Foggia, 71122 Foggia, Italy; lorenzo.lomuzio@unifg.it

**Keywords:** systemic lupus erythematosus, Epstein–Barr virus, oral lesions

## Abstract

In order to investigate the possible role of Epstein–Barr virus (EBV) in systemic lupus erythematosus (SLE) and its associated oral lesions, a pilot case–control study was performed. A total of 31 patients (18 females and 13 males) were enrolled in the study and divided into two groups: group A included 16 patients with diagnosis of SLE and group B included 15 healthy individuals. Salivary swab samples were collected and subjected to molecular screening by real-time quantitative PCR (qPCR) for the detection of EBV DNA. EBV DNA was significantly detected in 8/16 (50%) SLE patients and in 5/7 (71.4%) subjects with SLE-associated oral lesions. Since EBV is one of the most common viruses in the human population, it is difficult to understand if it is the causative agent of SLE or, vice versa, if SLE is able to trigger the reactivation of EBV. This study highlights a significant association between the presence of EBV and both SLE and SLE-related oral lesions and provides rationale for further investigation into the role of EBV in SLE pathogenesis.

## 1. Introduction

Systemic lupus erythematosus (SLE) is a chronic autoimmune rheumatic disease with unknown and unclear etiopathogenesis, with various clinical presentations, some of which are potentially life threatening [1,2,3].

SLE is characterized by B and T cell dysregulation [4] and autoantibody production leading to the potential attack of any organ system [1].

Skin and joints are the most commonly affected organs; however, kidneys with lupus nephritis, hearts with myocarditis and pericarditis, central nervous systems with cerebrovascular disease, retinae with nervous injury and potential loss of vision, muscles and lungs can also be involved [1].

SLE has a worldwide prevalence ranging between 12 and 50 per 100,000 individuals, with the highest incidence in individuals 40 years old and in females. The etiopathogenesis of SLE is still unclear, and it is characterized by a strong genetic predisposition, with specific haplotypes of the human leukocyte antigen (HLA) region, i.e., HLA-DRB1 (DRB1*1501 and DRB1*0301), and a concomitant co-factor that triggers the autoimmune response [5,6].

Many co-factors have been associated with the onset of SLE, including hormones, medications, UV exposure and infectious pathogens. Among the infectious pathogens, a possible candidate is Epstein–Barr virus (EBV), also referred to as human gammaherpesvirus 4, a double-stranded DNA (dsDNA) virus in the family Herpesviridae. EBV is endemic in the worldwide population because of its transmission modalities and its ability to integrate into the B cell genome, resulting in persistent infections that are common to most herpesviruses [7]. Interacting with the host immune system through several mechanisms, such as structural or functional molecular mimicry [8], superantigen production [9], bystander activation [10] and epigenetic factors [11,12], EBV can cause loss of tolerance, production of autoantibodies, tissue deposition of immune complexes and consequent tissue damage [13]. These mechanisms of virus-induced autoimmunity are also involved in the pathogenesis of SLE [13]. In addition, compared with healthy subjects, SLE patients are characterized by higher EBV load, anti-EBV antibody titers and abnormal expression of viral genes [5,14,15,16,17].

EBV is transmitted by saliva and initially targets the epithelial cells of the oropharynx, nasopharynx and tonsillar regions for its transcription and replication, and it subsequently infects B lymphocytes by specific envelope glycoproteins. After primary infection, EBV enters into a latent phase and lurks in resting memory B cells, where EBV replication is controlled by host immunity. Immune impairment can trigger EBV reactivation with a new lithic phase and production of infectious virions. This process initiates a new round of epithelial infection in the tonsils and viral shedding in the saliva [18].

Many studies to date have demonstrated an increased reactivation of EBV in SLE patients likely due to dysregulation of the latent phase or an enhanced switch to the lytic phase [19]. High levels of the viral transcription factor Zta, also called EB1, BZLF1 or ZEBRA, involved in the transition from the latent to the lytic phase are detectable in SLE patients [20]. Zta induces the transcription of viral and cellular genes and increases the production of autoantibodies against dsDNA (anti-dsDNA) [20]. Moreover, EBV encodes several viral homologues of human proteins, such as EBV IL-10 (vIL-10), which is homologous to the human IL-10 (hIL-10); it is produced during the lytic phase of the virus and encoded by the viral BCRF1 gene [18,20]. It is an immunosuppressive cytokine that inhibits IFN-γ and TNF-α production and CD8+ cytotoxic T cell and MHC-I expression in favoring immune surveillance inhibition [18,21]. Significantly high levels of vIL-10 have been observed in SLE patients’ plasma compared to unaffected controls, supporting the hypothesis of EBV reactivation in SLE patients [18]. Furthermore, vIL-10 levels in [18] correlated with levels of IgA antibodies specific to the EBV viral capsid antigen, which is an indirect measure of viral reactivation. In SLE patients, the anti-inflammatory effects of hIL-10 are overcome by vIL-10, resulting in an increase in inflammatory gene expression and an exacerbation of autoimmune responses [18].

Moreover, EBV nuclear antigens 1 and 2 (EBNA-1 and EBNA-2), which are able to prime the replication of EBV DNA, share highly similar structures with two lupus autoantigens, Sm B/B’ and Sm D1, respectively.

These autoantigens are nuclear proteins that constitute the common core of small nuclear ribonucleoprotein (snRNP) particles and are frequently targeted by anti-nuclear antibodies in SLE patients. The molecular mimicry between these lupus autoantigens and viral proteins may play a potential role in the induction and maintenance of the autoimmune response in SLE patients [22].

Furthermore, EBNA2 can bind SLE risk loci in gene regulatory regions, such as NFkB subunits, resulting in a downstream alteration in gene expression and an increased risk of developing SLE and other inflammatory diseases, which further suggests a possible causal role of EBV for SLE [23].

In addition, during the latent phase, EBV stimulates the proliferation of B lymphocytes and can immortalize host infected cells, as demonstrated in cell cultures. Persistent infection can mediate oncogenic growth of host cells with uncontrolled proliferation and immortalization [24]. This mechanism, along with the continuous production of viral proteins that can act as self-antigens and in specific conditions of genetic predisposition, can activate and sustain an autoimmune response such as in SLE [25].

Among the various SLE multi-organ clinical presentations, ulcerative oral lesions (7–41%) as well as periodontitis (40%) are the second most common muco-cutaneous manifestations in this disease and, interestingly, present similarities with EBV-related oral lesions [1,2,3,26,27].

The hypothesis of an etiopathogenetic correlation between EBV infection and SLE has long been suggested, although it has not been fully explained. Based on the observation of a high prevalence of antibodies to EBV antigens and EBV DNA in SLE patients, we hypothesized a possible increase in oral manifestations in SLE patients when EBV is in a replicative phase, because viral replication and autoimmune disease determine the same oral manifestations.

Detection of EBV in blood/serum is the most common used assay, although it is not possible to distinguish an active lytic phase with viral replication from a latent phase with a quiescent virus. On the contrary, the detection of EBV in saliva would be more consistent with active viral replication.

In the present study, the presence of EBV in saliva of SLE patients, compared with the control group of healthy patients, was investigated. Subsequently, the prevalence of oral lesions in both the study groups and the possible association between EBV and oral manifestations were evaluated.

## 2. Materials and Methods

### 2.1. Sample Collection

A total of 31 patients (18 females and 13 males, median age = 32 years (y)) were enrolled in the Unit of Internal Medicine “Guido Baccelli”, Department of Biomedical Sciences and Human Oncology, University of Bari Medical School, Bari, Italy, and divided in two groups on the basis of a presumptive diagnosis of SLE. Group A included 16 patients with diagnosis of SLE (13 females and 3 males, mean age = 39 y) whilst group B included 15 healthy individuals (5 females and 10 males, median age = 31 y). Salivary swab samples were collected from every patient by rubbing lining and keratinized mucosa with dry sterile swabs. Swabs were immediately stored at −20 °C until use. Each patient compiled an anamnestic questionnaire and was orally inspected for mucosal lesions.

The inclusion criteria for patients enrolled in the study were the presence of oral lesions, i.e., ulcers or white plaques with central erythema on keratinized and lining mucosa, and periodontitis diagnosed according to the Classification of American Academy of Periodontology of 2018, with interdental clinical attachment loss (CAL) measurements, probing depth (PD) measurements and radiographic bone loss measurements. For radiographic measurements, periapical radiographies no older than six months and in the possession of the patients were used.

The patients involved in the research signed a formal written informed consent form.

### 2.2. DNA Extraction

Swabs were immersed in 1 mL of viral transport medium consisting of Dulbecco’s modified Eagle’s medium (DMEM) supplemented with 5% fetal calf serum (FCS), 1000 IU/mL penicillin, 1000 μg/mL streptomycin and 10 μg/mL amphotericin B. Aliquots of the oropharyngeal swab extracts were combined, and, subsequently, 200 μL of each homogenate sample was used for DNA extraction by means of QIAamp cador Pathogen Mini Kit (Qiagen S.p.A., Milan, Italy), following the manufacturer’s protocol. The nucleic acid templates were stored at −80 °C until their use.

### 2.3. Analysis with Real-Time Quantitative PCR

Viral load in the positive samples was determined using real-time quantitative PCR (qPCR) assay, as previously described [16]. Ten microliters of DNA was added to the 15 μL reaction master mix (IQ™ Supermix, Bio-Rad Laboratories Srl) containing 0.9 μM of primers and 0.2 μM of probe. Forward 5EBT (5’-TCAACCTCTTCCATGTCACTGAGA-3’) and reverse 3EBT (5’-TGGGTGAGCGGAGGTTAGTAA-3’) primers amplified a 109-bp fragment of the highly conserved BLLF1 gene of EBV. The fluorogenic probe EBpr (5’-TCAGCCCCTCCACCAGTGACAATTC-3’), located between the PCR primers, contained a fluorescent reporter dye (6-carboxyfluorescein) at the 5‘ end and a fluorescent quencher dye (6-carboxy-tetramethyl-rhodamine) at the 3’ end. Thermal cycling consisted of activation of iTaq DNA polymerase at 95 °C for 10 min, 40 cycles of denaturation at 95 °C for 15 s and annealing extension at 60 °C for 1 min. EBV DNA copy numbers were calculated on the basis of standard curves generated by 10-fold dilutions of a plasmid containing the 109 bp fragment of the highly conserved BLLF1 gene of EBV.

### 2.4. Statistical Analysis

The quantitative variable (age) was defined as a new dummy variable (<32 y vs. ≥32 y) on the basis of the median age of patient. Qualitative and dummy variables were summarized as counts, and comparisons between independent groups were assessed by the chi-square test. Odds ratio (OR) and 95% confidence interval (95% CI) were calculated for each comparison.

Statistical analyses were performed using the freely available online tool EZR [28] for personal computers. A *p*-value < 0.05 was considered for statistical significance.

## 3. Results

In order to decipher the possible role of EBV in SLE-associated oral lesions, 16 SLE patients and 15 healthy individuals were enrolled in a pilot case–control study. Molecular screening by qPCR detected EBV DNA in a total of 9/31 (29%) salivary swabs. The viral loads of the positive samples ranged from 4.4 × 101 to 1.5 × 106 DNA copies/10 μL (mean 1.7 × 105 DNA copies; median 2.1 × 103 DNA copies). EBV DNA was detected in 8/16 (50%) SLE patients (group A) and in 1/15 (6.7%) asymptomatic patients (group B) (Table 1). When comparing patients from group A and B, a statistically very significant difference was observed in terms of EBV prevalence (*p* = 0.008, OR = 14.0 % CI = (1.5, 133.2)) (Table 1).

Moreover, the presence of EBV DNA in the salivary samples of patients with (5/7, 71.4%) and without (4/24, 16.7%) oral lesions was statistically very significant (*p* = 0.005, OR = 12.5, 95% CI = (1.8, 88.7)) (Table 2).

When comparing the presence of oral lesions in patients of group A (6/16, 37.5%) and group B (1/15, 6.7%) a significant difference (*p* = 0.04), was observed, although without significance in 95% CI (0.9, 81.1) related to OR (8.4) (Table 3).

Group A and B were also reanalyzed by age-based and sex cohorts of patients. Two out of 14 (14.3%) patients in the <32 y group and 7/17 (41.2%) patients in the ≥32 y group tested positive for EBV. However, this difference was not statistically significant (*p* = 0.11, OR = 0.3, 95% CI = (0.1, 1.3)). Viral DNA was detected in 2/8 (25%) patients in the male group and in 7/23 (30.4%) patients in the female group, without any statistical difference (*p* = 0.3, OR = 0.5, 95% CI = [0.1, 2.4]).

Group A was reanalyzed assessing clinical and hematological parameters of SLE with Epstein–Barr virus status (positivity or negativity) in salivary samples. Relevant differences between EBV+ and EBV− were not observed (Table 4).

## 4. Discussion

Many studies have investigated the possibility of an association between SLE and EBV infection [5,25,29,30,31,32,33,34]. Repeated reactivation of EBV that persists in a latent form in memory B cells of previously infected patients can activate autoreactive lymphocytes causing SLE disease flares. EBV DNA positivity and a higher sero-prevalence of EBV antibodies in SLE patients compared with healthy subjects confirmed EBV reactivation [5,14,15,16,17].

Our study extends the knowledge about a possible implication of EBV in SLE etiopathogenesis. Unlike previous studies, since detection of EBV in saliva is consistent with active viral replication, we used saliva as a preferential sample. By screening oral swabs, we detected EBV DNA in 8/16 (50%) SLE patients and only in 1/15 (6.6%) healthy subjects. Interestingly, we observed that most of the SLE patients with EBV DNA positivity presented oral lesions. A higher frequency (37.5%) of ulcerative oral lesions and/or periodontitis in SLE patients rather than in healthy subjects (6.66%) was observed, and a statistically significant difference was found between the two groups. No additional statistically significant difference emerged from our analysis of the results by age-based and sex cohorts.

Other previous studies have shown that herpesvirus DNA in patients with oral lesions [35] and persistent EBV infection can result in oral manifestations, oral hairy leukoplakia and EBV-positive mucocutaneous ulcers that are evocative of SLE-related ulcerative lesions, chiefly in immunocompromised patients [2].

Furthermore, the second most common mucocutaneous manifestations in SLE patients after butterfly rash are ulcerative oral lesions on keratinized mucosa or in lining mucosa (the soft palate, buccal and labial mucosa) [2,36]. The presence of ulcerations on keratinized mucosa, especially on the hard palate, should always be suspected given that common oral ulcerative lesions are observable only on lining mucosa [37]. Lesions on the hard palate are an acute sign that appears when SLE is in an active phase, and can sometimes be the only mucocutaneous manifestation. Early palatal lesions may appear as a red, round bleeding area, which progressively evolves into large erythematous patches [38]. On the soft palate, buccal and labial mucosa, it is possible to observe atrophic white plaques with central erythema, white radiating keratotic striae and a peripheral telangiectasia, eventually evolving into an atrophic lesion with a keratotic border [38]. Other SLE-related oral clinical manifestations, which usually occur during active disease, are aphthous-like ulcers generally present with multiple lesions of less than 1 cm in diameter, characterized by a white to yellow coating and a surrounding red rim, and cheilitis with crusty, painful ulcers that often affect the vermilion zone of the lower lip [37]. Moreover, periodontitis is frequently reported in these patients. Periodontitis has a bacterial etiopathogenesis but SLE may act as a possible risk factor, favoring the inflammatory status, changes in innate immune system and a decrease in oral hygiene because of the pain from lesions during teeth brushing [3,27,39]. Detection of EBV in advanced types of periodontal and endodontic diseases indicates a synergic action of EBV with periodontal disease-associated microbial biofilms [27,40].

Although EBV is common in the human population, it is difficult to understand if EBV is the causative agent of SLE or, vice versa, if SLE is able to trigger reactivation of EBV by immunodysregulative mechanisms. However, our study supports the hypothesis about a close association between EBV infection and SLE disease and SLE-associated oral lesions, and it favors an early detection of EBV infection in SLE patients. Dentists observing suspected chronic ulcerative lesions at oral inspection of patients, especially on the hard palate, keratinized mucosa or tongue, should suspect SLE and/or a possible manifestation of persistent EBV infection. These patients should be suggested to undergo diagnostic investigations to confirm or rule out SLE.

Furthermore, good hygiene practices and biosafety measures, including testing for EBV in oral swabs, should be adopted by SLE patients with oral lesions to prevent transmission of EBV in households or to persons in close contact.

## Figures and Tables

**Table 1 jcm-10-04995-t001:** Epstein–Barr virus (EBV) detection by qPCR in salivary swabs from patients with systemic lupus erythematosus (SLE+) and control subjects (SLE−).

Group	*n*. Subjects	EBV+ (%)	EBV− (%)	*p* Value	OR	CI 95%
A SLE+	16	8 (50)	8 (50)	0.008 **	14.0	[1.5, 133.2]
B SLE−	15	1 (6.7)	14 (93.3)			
Total	31	9	22			

+: positive; −: negative; OR: odds ratio; CI95% confidence interval 95%; ** very significant.

**Table 2 jcm-10-04995-t002:** Epstein–Barr virus (EBV) detection by qPCR in salivary swabs from subjects with oral lesions (OL+) and without oral lesions (OL−).

	*n*. Subjects	EBV+ (%)	EBV− (%)	*p* Value	OR	CI 95%
OL +	7	5 (71.4)	2 (28.6)	0.005 **	12.5	[1.8, 88.7]
OL −	24	4 (16.7)	20 (83.3)			
total	31	9	22			

+: positive; −: negative; OR: odds ratio; CI 95% confidence interval 95%; ** very significant.

**Table 3 jcm-10-04995-t003:** Oral lesions (OL) in patients with systemic lupus erythematosus (SLE+) and control subjects (SLE−).

Group	*n*. Subjects	OL+ (%)	OL− (%)	*p* Value	OR	CI 95%
A SLE+	16	6 (37.5)	10 (62.5)	0.04 *	8.4	[0.9, 81.1]
B SLE−	15	1 (6.7)	14 (93.3)			
total	31	7	24			

+: positive; −: negative; OR: odds ratio; CI 95% confidence interval 95%; * significant.

**Table 4 jcm-10-04995-t004:** Frequency of clinical and hematological parameters in patients affected by systemic lupus erythematosus (SLE) (group A) tested positive or negative for Epstein–Barr virus (EBV) in salivary samples.

Clinical and Hematological Parameters	EBV+ (%)	EBV− (%)	Total (%)
• **Fever**	0/8 (0)	0/8 (0)	0/16 (0)
• **Mucocutaneous involvement**			
Inflammatory Lupus malar rash	1/8 (12.5)	2/8 (25)	3/16 (18.7)
Alopecia	1/8 (12.5)	0/8 (0)	1/16 (6.25)
Mucosal ulcers	4/8 (50)	2/8 (25)	6/16 (37.5)
Cutaneous vasculitis (ulcers, tender finger nodules, periungual hemorrhages)	3/8 (37.5)	3/8 (37.5)	6/16 (37.5)
• **Articular involvement and/or Arthritis**	3/8 (37.5)	5/8 (62.5)	8/16 (50)
• **Myositis**	0/8 (0)	0/8 (0)	0/16 (0)
• **Renal involvement**			
Urinary casts	0/8 (0)	0/8 (0)	0/16 (0)
Proteinuria	1/8 (12.5)	1/8 (12.5)	2/16 (12.5)
Hematuria	0/8 (0)	0/8 (0)	0/16 (0)
• **Pleurisy**	1/8 (12.5)	2/8 (25)	3/16 (18.7)
• **Pericarditis and/or myocarditis**	0/8 (0)	0/8 (0)	0/16 (0)
• **Neuropsychiatric disorders**			
Seizure	0/8 (0)	0/8 (0)	0/16 (0)
Psychosis	0/8 (0)	0/8 (0)	0/16 (0)
Headache	0/8 (0)	0/8 (0)	0/16 (0)
Cranial nerve disorders	0/8 (0)	0/8 (0)	0/16 (0)
Visual disturbance	0/8 (0)	0/8 (0)	0/16 (0)
Organic brain syndrome	0/8 (0)	0/8 (0)	0/16 (0)
Cerebrovascular accidents	0/8 (0)	0/8 (0)	0/16 (0)
• **Hematopoietic involvement**			
Leukopenia	0/8 (0)	1/8 (12.5)	1/16 (6.25)
Thrombocytopenia	1/8 (12.5)	1/8 (12.5)	2/16 (12.5)
Anemia	2/8 (25)	4/8 (50)	6/16 (37.5)
• **Low complement (C3 and/or C4)**	4/8 (50)	5/8 (62.5)	9/16 (56.2)
• **Antinuclear Antibodies (ANA)**	7/8 (87.5)	7/8 (87.5)	14/16 (87.5)
• **Anti-dsDNA antibodies**	5/8 (62.5)	6/8 (75)	11/16 (68.7)
• **Anti-phospholipid antibodies**	4/8 (50)	2/8 (25)	6/16 (37.5)

+: positive; −: negative.

## Data Availability

Data is contained within the article.

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
