# Peer review of "Epstein–Barr Virus in Salivary Samples from Systemic Lupus Erythematosus Patients with Oral Lesions"

_jcm, 2021, doi:10.3390/jcm10214995_

Round 1

Reviewer 1 Report

Table 1 should describe the frequency of clinical and hematological parameters in the analyzed patients. This is to understand the target cases of this study, and there is no point in listing the common symptoms of SLE in this manuscript.

Author Response

R1.1 Table 1 should describe the frequency of clinical and hematological parameters in the analyzed patients. This is to understand the target cases of this study, and there is no point in listing the common symptoms of SLE in this manuscript.

Reply to R1.1 We removed Table 1 from the manuscript and table 5 (table 4 in the revised version) was modified adding frequence data regarding clinical and hematological parameters in the analyzed patients, as requested.

Reviewer 2 Report

In this manuscript, Buonavoglia et al. investigated the association between the presence of EBV and SLE-related oral lesions. The findings are certainly of interest and appropriate to publish in the Journal of Clinical Medicine. The paper is clearly written and the experiments are nicely presented. However, this reviewer has a couple of specific queries.

  1. Since titers of antibodies against EBV antigens are elevated compared to healthy controls in SLE and this elevation is considered as one of the first symptoms, did the authors check that in saliva samples?
  2. Is it possible to detect the type of EBV strain in saliva samples? Please discuss.
  3. The tables are not well organized and hard to follow.
  4. Please remove the yellow marking in the result section (Line 212-214).

Author Response

In this manuscript, Buonavoglia et al. investigated the association between the presence of EBV and SLE-related oral lesions. The findings are certainly of interest and appropriate to publish in the Journal of Clinical Medicine. The paper is clearly written and the experiments are nicely presented. However, this reviewer has a couple of specific queries.

R2.1 Since titers of antibodies against EBV antigens are elevated compared to healthy controls in SLE and this elevation is considered as one of the first symptoms, did the authors check that in saliva samples?

Reply to R2.1 We thank the referee for the appropriate observation. However, we did not evaluate antibody titers against EBV in saliva samples as our aim was the evaluation of the presence and quantification of EBV genome in the saliva samples.

R2.2 Is it possible to detect the type of EBV strain in saliva samples? Please discuss.

Reply to R2.2 The aim of the manuscript was the evaluation of the presence and quantity of EBV genome in SLE-patients. Accordingly, a real time PCR test targeted to a highly conserved gene of EBV genome was performed on salivary samples. Therefore, strain characterization was not performed.

R2.3 The tables are not well organized and hard to follow.

Reply to R2.3 We agree that the tables needed a restyle in order to clarify information presented. Accordingly, in the revised version we removed table 1 (see reply to R1.1) and we modified table 4 (table 5 in the original version).

R2.4 Please remove the yellow marking in the result section (Line 212-214).

Reply to R2.4 This was done